# Affective bias as a rational response to the statistics of rewards and punishments

Erdem Pulcu[1], Michael Browning[1,2]*

[1]Department of Psychiatry, University of Oxford, Oxford, United Kingdom; [2]Oxford Health NHS Foundation Trust, Oxford, United Kingdom

**Abstract** Affective bias, the tendency to differentially prioritise the processing of negative relative to positive events, is commonly observed in clinical and non-clinical populations. However, why such biases develop is not known. Using a computational framework, we investigated whether affective biases may reflect individuals' estimates of the information content of negative relative to positive events. During a reinforcement learning task, the information content of positive and negative outcomes was manipulated independently by varying the volatility of their occurrence. Human participants altered the learning rates used for the outcomes selectively, preferentially learning from the most informative. This behaviour was associated with activity of the central norepinephrine system, estimated using pupilometry, for loss outcomes. Humans maintain independent estimates of the information content of distinct positive and negative outcomes which may bias their processing of affective events. Normalising affective biases using computationally inspired interventions may represent a novel approach to treatment development.

DOI: https://doi.org/10.7554/eLife.27879.001

## Introduction

When learning about and interacting with the world, individuals vary in the extent to which their beliefs and behaviours are influenced by the events they experience. Often this variation displays an affective gradient with some individuals being more influenced by positive and others by negative events. For example, many people display an optimism bias, updating their beliefs to a greater extent following positive than negative outcomes (*Sharot and Garrett, 2016*). The opposite effect, a tendency to be more influenced by negative events, has been argued to cause illnesses such as depression and anxiety (*Mathews and MacLeod, 2005*). However, relatively little work has explored why individuals might develop affective biases in the first place. This question is of particular importance as understanding the mechanisms which lead to the development of affective bias is an essential first step in the development of novel treatments designed to alter this process and thus reduce symptoms of depression and anxiety. One way of answering *why* individuals develop affective bias is to consider *when* affective biases might be the appropriate way to think about the world. In this study we draw on recent advances from the computational neuroscience of learning to investigate whether affective biases may be understood in terms of how informative an individual judges an event to be. Below we describe the conceptual framework of this proposal and then suggest how this may be used to account for the occurrence of affective biases.

Recent computational work has demonstrated that individuals' expectations are influenced more by those events which carry more information; that is, those events which improve predictions of future outcomes to a greater degree (*Behrens et al., 2007*; *Browning et al., 2015*; *MacKay, 2003*; *Nassar et al., 2012*). One factor which influences how informative an event is is the changeability, or volatility, of the underlying association which is being learned. For example, imagine trying to learn what your colleagues think about your performance at work, based solely on their day-to-day feedback. One colleague seems to have a stable positive view of you, complimenting you on your work

*For correspondence:
michael.browning@psych.ox.ac.uk

on 80% of the occasions you meet and never increasing or decreasing this frequency. In this case, each particular event (being complimented or not) provides little new information about what your colleague thinks about you, as you will always have an 80% chance of being complimented the next time you meet. In contrast, a second colleague's appraisal of you seems to be more changeable, with periods when they think highly of you and compliment you regularly and others when they rarely compliment you at all. In this case each event provides more information; if you have recently been complimented by this colleague it is more likely that their opinion of you is currently high and they will compliment you the next time you meet (*Figure 1B*). When learning what your colleagues currently think about you, you should be more influenced by whether the second, more volatile, colleague compliments you or not, because this provides more useful information than the behaviour of the stable colleague.

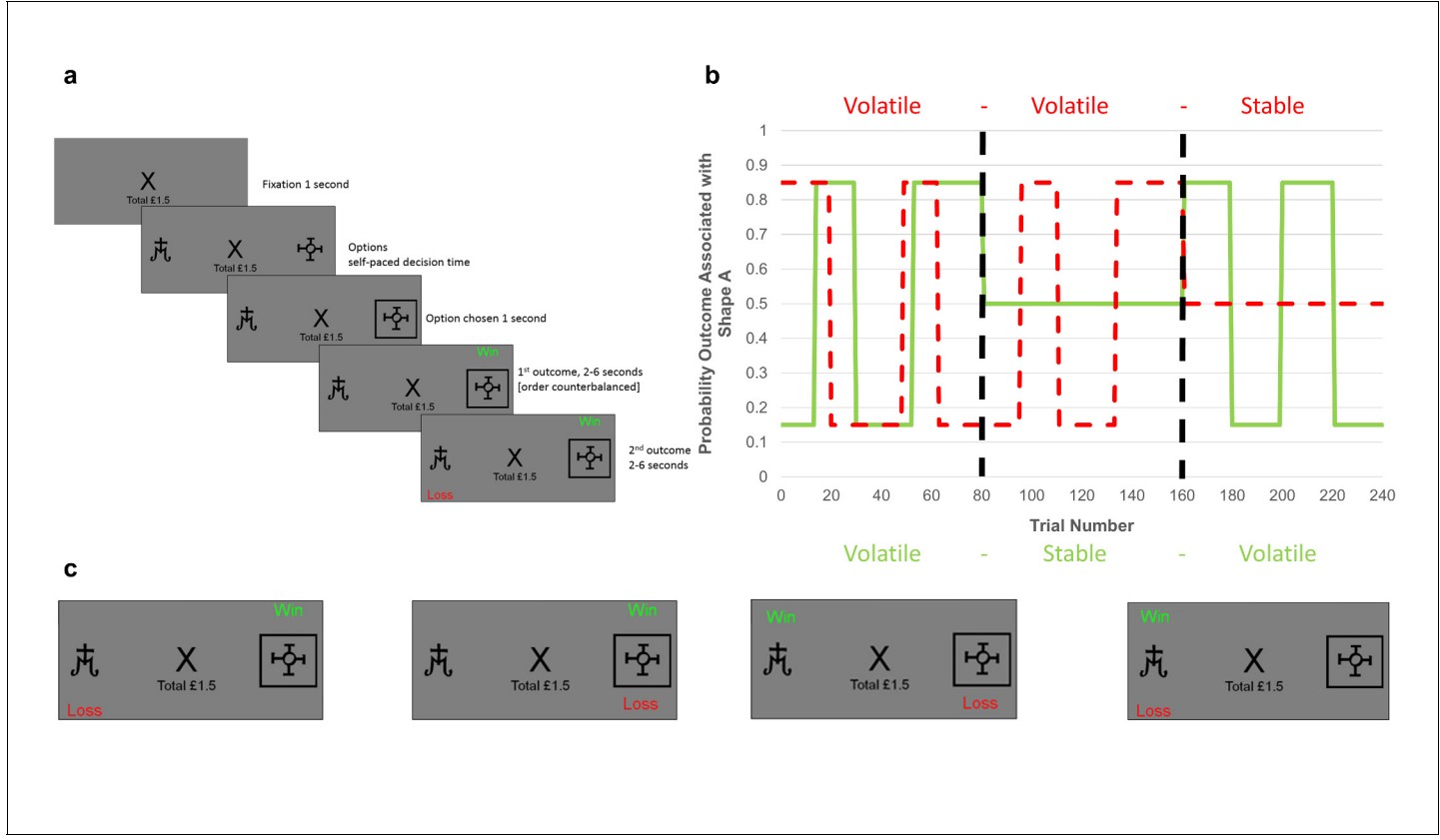

**Figure 1.** Task structure. (**A**) Timeline of one trial from the learning task used in this study. Participants were presented with two shapes (referred to as shape 'A' and 'B') and had to choose one. On each trial, one of the two shapes was be associated with a 'win' outcome (resulting in a win of 15 p) and one with a 'loss' outcome (resulting in a loss of 15 p). The two outcomes were independent, that is knowledge of the location of the win provided no information about the location of the loss (see description of panel C below). Using trial and error participants had to learn where the win and loss were likely to be found and use this information to guide their choice in order to maximise their monetary earnings. (**B**) Overall task structure. The task consisted of 3 blocks of 80 trials each (i.e. vertical, dashed, dark lines separate the blocks). The y-axis represents the probability, *p*, that an outcome (win in solid green or loss in dashed red) will be found under shape 'A' (the probability that it is under shape 'B' is 1-*p*). The blocks differed in how volatile (changeable) the outcome probabilities were. Within the first block both win and loss outcomes were volatile, in the second two blocks one outcome was volatile and the other stable (here wins are stable in the second block and losses stable in the third block). The volatility of the outcome influences how informative that outcome is. Consider the second block in which the losses are volatile and the wins stable. Here, regardless of whether the win is found under shape 'A' or shape 'B' on a trial, it will have the same chance of being under each shape in the following trials, so the position of a win in this block provides little information about the outcome of future trials. In contrast, if a loss is found under shape 'A', it is more likely to occur under this shape in future trials than if it is found under shape 'B'. Thus, for the second block losses provide more information than wins and participants are expected to learn more from them. (**C**) The four potential outcomes from a trial. Win and loss outcomes were independent, and so participants had to separately estimate where the win and where the loss would be on each trial in order to complete the task. This manipulation made it possible to independently manipulate the volatility of the two outcomes.

DOI: https://doi.org/10.7554/eLife.27879.002

Within a reinforcement learning framework, the influence of events on one's belief is captured by the learning rate parameter, with a higher learning rate reflecting a greater influence of more recently experienced events (*Sutton and Barto, 1998*). Humans adjust their learning rate precisely as described above, using a higher learning rate for events, such as those occurring in a volatile context, which they estimate to be more informative (*Behrens et al., 2007*; *Browning et al., 2015*; *Nassar et al., 2012*). The neural mechanism by which this modification of learning rate is achieved is thought to depend on activity of the central norepinepheric system (*Yu and Dayan, 2005*), with increased phasic activity of the system, which may be estimated using pupilometry (*Joshi et al., 2016*), reporting the occurrence of more informative events (*Browning et al., 2015*; *Nassar et al., 2012*) and acting to enhance the processing of these events (*Aston-Jones and Cohen, 2005*).

This computational framework provides an overarching logic for when an individual might develop an affective biases; individuals should bias their processing towards those affective events that they estimate to be most informative. As well as providing a novel reformulation of why affective biases may develop, this framework also suggests a potential novel method for modifying such biases; for example, if a higher estimate of the information content of negative relative to positive events leads to negative affective bias, interventions which redress these estimates should also reduce the negative bias.

However, a number of critical questions concerning this account remain outstanding. Firstly, no previous study has demonstrated that humans maintain separate estimates of the information content of positive and negative events. While a number of studies have examined the effect of volatility on learning (*Behrens et al., 2008*; *Behrens et al., 2007*; *Browning et al., 2015*; *Nassar et al., 2012*), they have all utilised only one type of outcome (i.e. rewards or punishments) and thus their results could be accounted for by learners maintaining an estimate of how volatile the general environment is and learning more rapidly to all outcomes in those environments they judge to be more volatile, rather than estimating the information content of specific outcomes. We tested whether these specific estimates were maintained using a novel learning task (*Figure 1*) in which participant choice led to both positive and negative outcomes, with the volatility of the outcomes (and therefore their information content) being independently manipulated in separate task blocks. Secondly, for estimates of the information content of positive and negative events to lead to the development of affective biases, the estimates themselves must be malleable. We assessed this malleability by testing whether the volatility manipulation described above altered participants' estimated information content, as reflected by the learning rates they used. Lastly, while activity of the central NE system has been argued to represent estimates of volatility, it is not clear whether or how this system might multiplex separate representations of the volatility of different classes of event, such as the positive and negative outcomes examined here. We investigated this using pupilometry as a measure of NE activity while participants completed the task. We hypothesised that humans maintain separable estimates of the information content of positive and negative outcomes, that we could measure and manipulate these estimates using our task and that phasic NE activity yoked to a specific type of outcome would track the volatility of that outcome.

## Results

30 participants (see *Table 1* for demographic information) completed a two option learning task in which, on every trial, one option would be associated with a monetary win and one with a loss (*Figure 1*). The win and loss outcomes occurred independently which required participants to learn separately which shape was associated with each outcome (*Figure 1c*). The information content of the outcomes was varied across the three blocks by altering the volatility of the stimulus-outcome associations (*Figure 1b*). We estimated separate learning rates for the positive and negative outcomes by fitting a computational model (see Materials and methods) to participant choice in each task block. This allowed us to test whether participants independently altered the learning rates they used for the win and loss outcomes in response to how informative that outcome was (i.e. its volatility). Pupilometry data was collected during the task as a measure of activity in the central NE system.

**Table 1.** Demographic details of participants

| Measure | Mean (SD) |
| --- | --- |
| Age | 30.52 (9.51) |
| Gender | 76% Female |
| QIDS-16 | 5.03 (3.95) |
| Trait-STAI | 35.79 (10.63) |

QIDS-16; Quick Inventory of Depressive Symptoms, 16 item self-report version. Trait-STAI; Speilberger State-Trait Anxiety Inventory, trait form. Note that scores of 6 or above on the QIDS-16 indicate the presence of depressive symptoms. The trait-STAI has no standard cut off scores.

DOI: https://doi.org/10.7554/eLife.27879.003

## Do human learners maintain independent estimates of the information content of positive and negative outcomes?

As predicted, participants' learning rates for positive and negative outcomes reflected the information content of the outcomes in the learning task (block volatility x parameter valence; $F_{(1,28)}$ =27.97, p<0.001; *Figure 2*). Specifically, learning rates were higher for win ($F_{(1,28)}$ =15.47, p=0.001) and loss ($F_{(1,28)}$ =18.02, p<0.001) outcomes when they were volatile (informative) than when they were stable (not informative). Similarly the learning rate for wins was higher than that for losses when wins were more volatile than losses ($F_{(1,28)}$ =26.02, p<0.001) and the learning rate for losses was higher than for wins when losses were more volatile ($F_{(1,28)}$ =6.74, p=0.015). These results demonstrate that participants maintain independent estimates of the information content of positive and negative outcomes and that it is possible to alter these estimates using a simple volatility manipulation. In contrast to the effects on learning rate there were no significant effects of the task on the inverse temperature parameter of the learning model (*Figure 2b*; $F_{(1,28)}$ =0.01, p=0.92) indicating that, as intended, the volatility manipulation specifically altered learning rate. See the Figure Supplements for *Figure 2* for additional analysis of the behavioural results as well as an additional experiment in which the impact of expected uncertainty was assessed.

## Does activity of the central NE system, as Estimated by Pupil Dilation, Track the Volatility of Positive and Negative Outcomes?

Next, we investigated the extent to which central NE activity, as estimated using pupilometry, was related to the information content of positive and negative outcomes in the learning task. Consistent with the behavioural findings a significant interaction between block volatility and outcome valence was found for the degree to which participants' pupils dilated in response to outcome receipt (*Figure 3*; $F_{(1,27)}$=6.16; p=0.02). In other words, participants' pupils dilated more on receipt of an outcome when that outcome was volatile (informative) relative to when it was stable (not informative). This effect was not further modified by the time bin following outcome (block volatility x outcome valence x time; $F_{(5,135)}$=1.13, p=0.35). Analysing the positive and negative outcomes separately indicated that the effect of block volatility was significant for the loss outcomes ($F_{(1,27)}$=10.46, p=0.003), but not for the win outcomes ($F_{(1,27)}$=0.38, p=0.54). Indeed a direct statistical comparison of the size of the volatility effect between the positive and negative outcomes indicated a greater effect of volatility on the negative relative to positive outcomes (outcome volatility x valance; $F_{(1,27)}$ =4.34, p=0.047). This effect was seen on the background of a generally greater pupil dilation to receipt of a loss relative to a win (main effect of valence; $F_{(1,27)}$=16.7, p<0.001).

## Are the behavioural and pupilometry measures capturing the same process?

As central NE activity is thought to mediate the effect of outcome information content on participant choice (*Yu and Dayan, 2005*), there should be a relationship between how much a participant's pupils differentially dilate in response to an outcome during the informative and non-informative blocks and the degree to which that participant adjusts their learning rate between blocks for the same outcome. We tested this by assessing the correlation between the change in mean pupil response between blocks and the change in behaviourally estimated learning rates, separately for

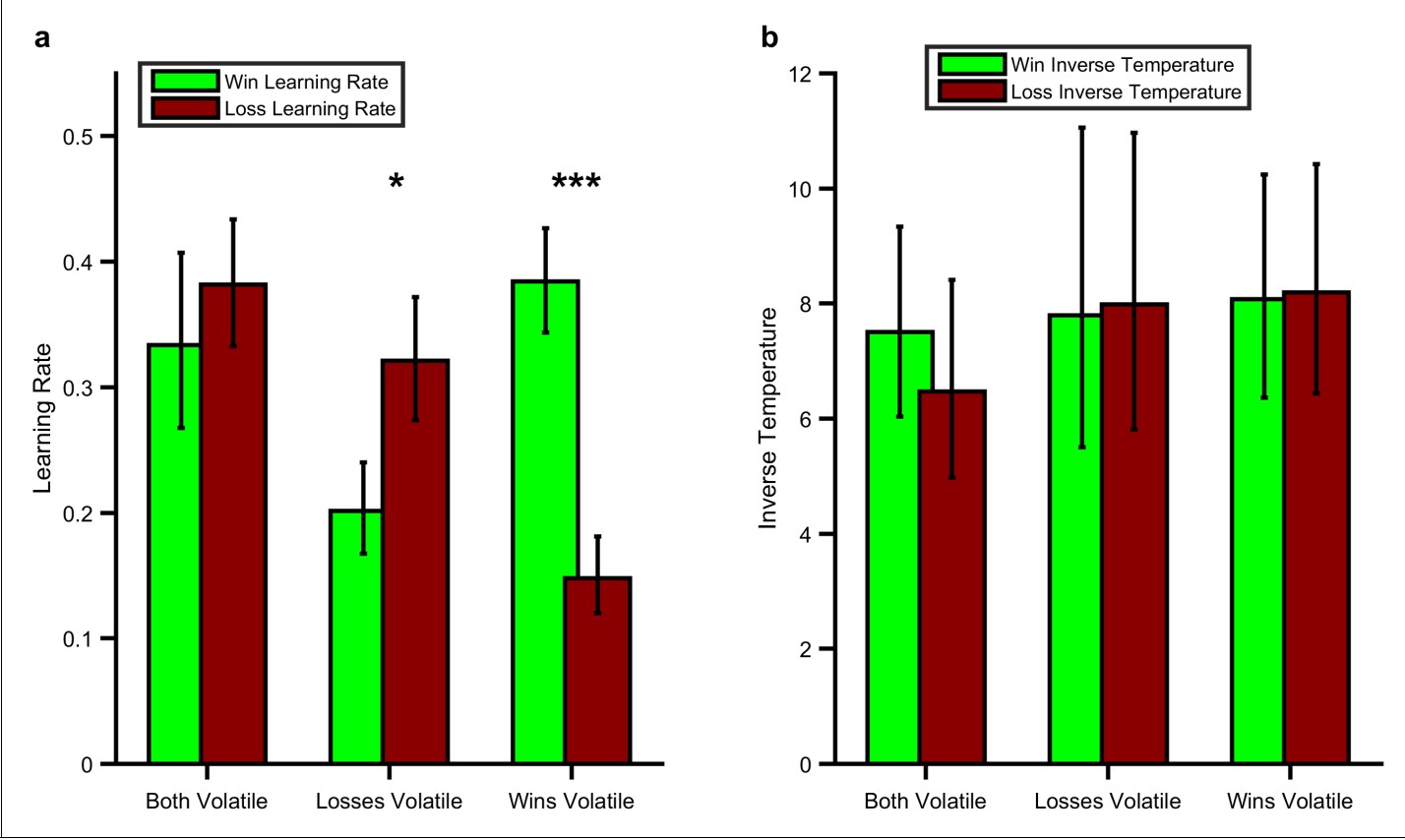

**Figure 2.** Effect of the Volatility Manipulation on Participant Behaviour. (**A**) Mean (SEM) learning rates for each block of the learning task. As can be seen the win learning rates (light green bars) and loss learning rate (dark red bars) varied independently as a function of the volatility of the relevant outcome $F_{(1,28)}$ =27.97, p<0.001, with a higher learning rate being used when the outcome was volatile than stable (*p<0.05, ***p<0.001 for pairwise comparisons). (**B**) No effect of volatility was observed for the inverse temperature parameters ($F_{(1,28)}$ =0.01, p=0.92). Source data available as *Figure 2—source data 1*. See *Figure 2—figure supplement 1* for an analysis of this behavioural effect which does not rely on formal modelling and *Figure 2—figure supplement 2* for an additional task which examines the behavioural effect of expected uncertainty.
DOI: https://doi.org/10.7554/eLife.27879.004

The following source data and figure supplements are available for figure 2:

**Source data 1.** Fitted model parameters, questionnaire measures and mean pupil response to volatility for all participants.
DOI: https://doi.org/10.7554/eLife.27879.007
**Figure supplement 1.** Analysis of switching behaviour in the learning task.
DOI: https://doi.org/10.7554/eLife.27879.005
**Figure supplement 2.** Magnitude task.
DOI: https://doi.org/10.7554/eLife.27879.006

wins and losses. As can be seen (*Figure 4*) the change in pupil response to loss outcomes between blocks was significantly correlated with the change in loss learning rate (r(28)=0.5, p=0.009) but pupil response to win outcomes was not correlated with change in win learning rate (r(28)=-0.08, p=0.7). This correlation was significantly greater for losses than for wins (Fisher r-to-z transformation z = 2.27, p=0.02).

## Discussion

Humans adapt the degree to which they are influenced by a positive and negative outcome in response to how informative they estimate those outcomes to be. These estimates produce a bias resulting in preferential learning, with a higher learning rate being used for the outcome which is most informative. These estimates are also malleable and thus may represent one route by which

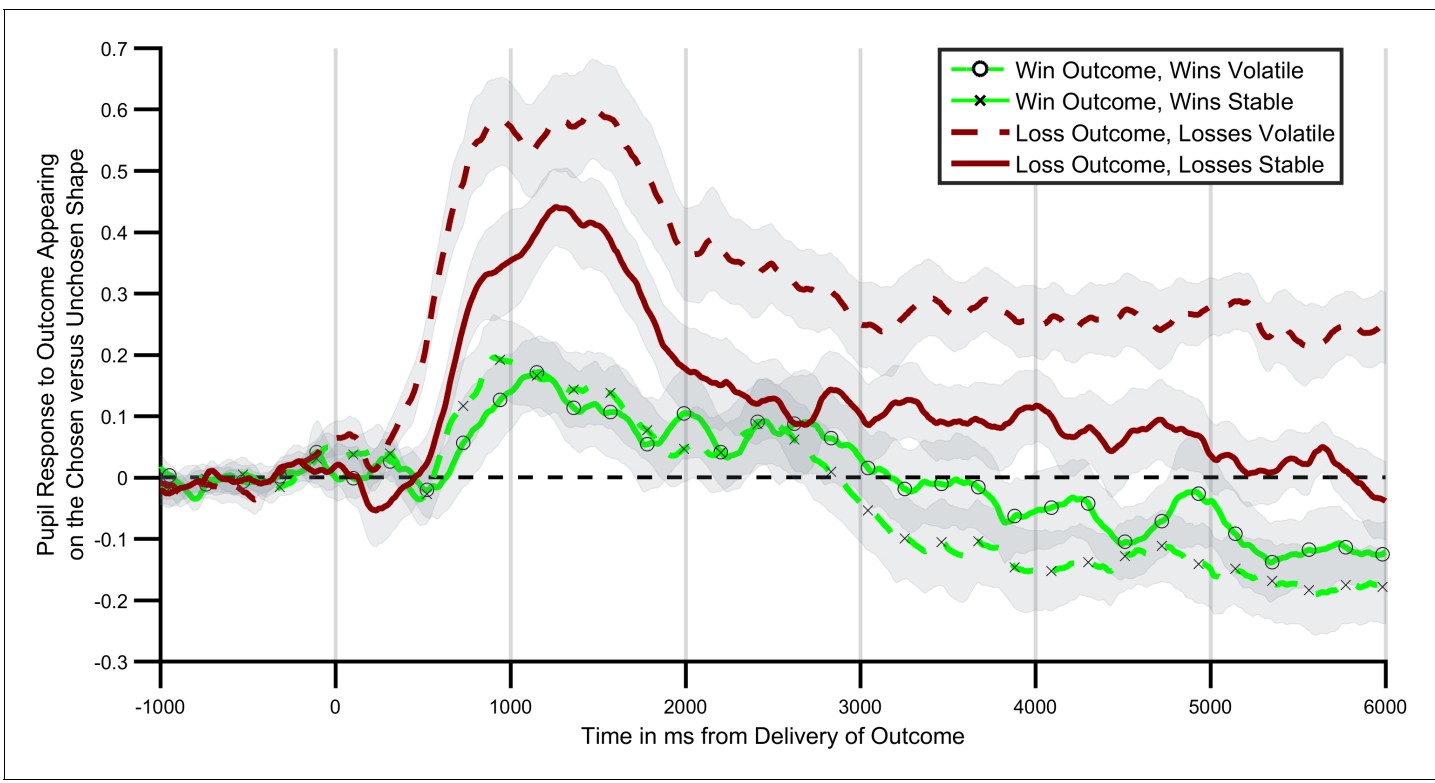

**Figure 3.** Pupil response to outcome delivery during the learning task. Lines illustrate the mean pupil dilation to an outcome when it appears on the chosen relative to the unchosen shape, across the 6 s after outcomes were presented. Light green lines (with crosses and circles) report response to win outcomes, dark red lines report response to loss outcomes. Solid lines report blocks in which the wins were more informative (volatile), dashed lines blocks in which losses were more informative. As can be seen pupils dilated more when the relevant outcome was more informative, with this effect being particularly marked for loss outcomes. Shaded regions represent the SEM. *Figure 3—figure supplement 1* plots the timecourses for trials in which outcomes were or were not obtained separately, and *Figure 3—figure supplement 2* reports the results of a complimentary regression analysis of the pupil data.

DOI: https://doi.org/10.7554/eLife.27879.008

The following figure supplements are available for figure 3:

**Figure supplement 1.** Individual time courses for trials in which wins (panel **a**) and losses (panel **b**) are either received or not received.

DOI: https://doi.org/10.7554/eLife.27879.009

**Figure supplement 2.** Regression analysis of pupil data.

DOI: https://doi.org/10.7554/eLife.27879.010

affective biases develop. A physiological measure of central NE activity was associated with this process, although this was only seen convincingly for loss outcomes.

Previous work has demonstrated that humans adapt their learning in response to subtle statistical aspects of the environment, such as employing an increased learning rate in volatile, or changeable, contexts (*Behrens et al., 2007*; *Browning et al., 2015*; *Nassar et al., 2012*). These previous findings could be explained by learners maintaining an estimate of how volatile the general environment is and learning more rapidly from all outcomes experienced in those environments they judge to be more volatile. However, the results of the current study, in which learning rates for positive and negative outcomes were seen to be altered independently, cannot be accounted for by a general estimate of environmental volatility. Rather, this behaviour requires the parallel representation of the estimated volatility of distinct outcomes which are then used to specifically tune the learning from that outcome. More generally these results suggest that human learners are able to maintain independent estimates of the information content of different events and use these estimates to rationally adjust their learning, in this case producing a valence dependent affective bias.

In the current study we investigated the link between the learning rate used by participants, which provides a behavioural index of how informative they estimate an outcome to be, and pupil

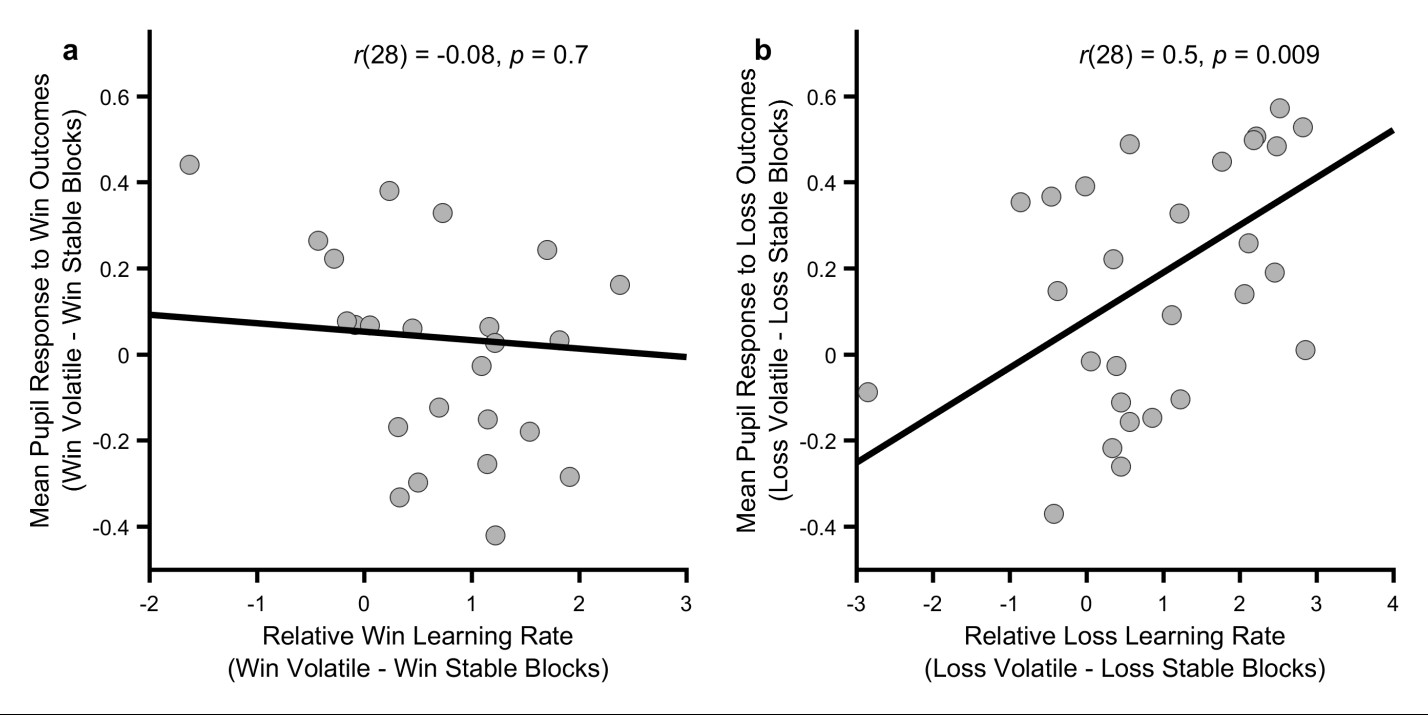

**Figure 4.** Relationship between behavioural and physiological measures. The more an individual altered their loss learning rate between blocks, the more that individual's pupil dilation in response to loss outcomes differed between the blocks (panel **b**; p=0.009), however no such relationship was observed for the win outcomes (panel **a**; p=0.7). Note that learning rates are transformed onto the real line using an inverse logit transform before their difference is calculated and thus the difference score may be greater than ±1. *Figure 4—figure supplements 1* and *2* describe the relationship between these measures and baseline symptoms of anxiety and depression.

DOI: https://doi.org/10.7554/eLife.27879.011

The following figure supplements are available for figure 4:

**Figure supplement 1.** Relationship between symptom scores and behavioural adaptation to volatility.
DOI: https://doi.org/10.7554/eLife.27879.012

**Figure supplement 2.** Relationship between symptom scores and pupillary adaptation to volatility.
DOI: https://doi.org/10.7554/eLife.27879.013

dilation which has been shown to correlate with central norepinepheric activity (*Joshi et al., 2016*). Pupil dilation in response to outcome receipt differed as a function of the information content of the outcome, although this was only significant for losses. Specifically, when losses were informative, the difference in pupil dilation between trials in which a loss was received and when it was not received was greater than when the losses were not informative. This result is similar to previously reported findings of an increased pupil response to outcomes in a volatile context (*Browning et al., 2015*; *Nassar et al., 2012*), although these earlier studies reported a general increase in pupil dilation rather than a dilation conditioned on receipt of the outcome. A possible explanation of this difference is that, as discussed above, in order to complete the tasks used in previous studies, which involved only one class of outcome, only an estimate of the general environmental volatility is required, whereas to perform the current task a volatility signal dependent on the outcome presented is needed. In other words, the volatility signal found in the pupil data from the current study is of the form required for participants to accurately perform the task. This suggests a degree of flexibility of the pupillary volatility signal, in that it may reflect the general volatility of a learned association or the volatility of specific dimensions of more complex associations depending on task demands. It is not clear whether these general and specific volatility signals are produced by a single or separate neural systems, although it may be possible to address this question using a task in which the total volatility of all task outcomes is manipulated independently of the volatility of individual outcomes.

The effect of outcome volatility on pupil dilation in the current study was significantly greater for loss than win outcomes with the correlation between this signal and behaviour also being significantly greater for losses. This surprising result raises the possibility that central norepinepheric activity is particularly related to the information content of negative, as opposed to positive outcomes. However, while we are not aware of previous studies which have reported pupillary volatility signals to positive and negative outcomes from a single task as in the current study, previous work has reported the presence of pupillary volatility signals in reward only tasks (*Nassar et al., 2012*). This suggests that the norepinpeheric system does respond to the volatility of positive outcomes, but that this response is less pronounced than that for negative outcomes. One explanation for this may be that, as discussed above, the volatility signal in the current task modified pupillary response to outcome receipt vs. non-receipt. The receipt of a loss led to a significantly greater pupil dilation than that produced by a win (see *Figure 3—figure supplement 1*) and thus the volatility effect, which modifies the relative dilation observed when an outcome is received, may be less apparent for wins. Of course, this explanation leaves open the question as to why the pupillary response to similar magnitude losses and wins is asymmetric in the first place. A similar pupillary asymmetry has been reported in decision making tasks designed to assess behavioural loss aversion (*Tversky and Kahneman, 1992*; *Yechiam and Telpaz, 2011*). It may be, therefore that the greater pupillary response to both the occurrence of a loss and to loss volatility is related to the general overweighting of loss relative to win outcomes reported in the broader decision making literature. If correct, this would suggest that increasing the relative magnitude of the pupillary response to the receipt of a win relative to a loss (for example, by increasing the salience of wins by increasing their magnitude) would also increase the size of the pupillary response to win volatility.

The pupilometry measure included in the current study raises the possibility that estimated information content may be influenced by pharmacological as well as cognitive interventions. Pupil size is influenced by the activity of a number of central neurotransmitters including norepinephrine (*Joshi et al., 2016*) and previous work exploring the neural systems which control response to volatility have predicted a key role for NE (*Yu and Dayan, 2005*) suggesting it as an obvious pharmacological target. A single study has reported an effect of atomoxetine, a norepinephrine reuptake inhibitor, on learning in a volatile environment (*Jepma et al., 2016*) although no previous work has examined the effect of a pharmacological intervention on learning to positive vs. negative outcomes. It would be interesting to test whether a pharmacological manipulation of norepinepheric function was able to modify the outcome specific volatility effect demonstrated in this paper as such an effect may indicate a clinically useful interaction between pharmacological and cognitive interventions. A pharmacological approach could also be used to investigate related mechanistic questions such as the greater pupillary response to loss than win outcomes and the greater pupillary signal for loss outcome volatility discussed above. Specifically, a greater impact of a pharmacological manipulation on learning rates for losses than wins would provide experimental evidence for a preferential role for the NE system in estimates of the information content of losses.

The parallel representation of estimated information content of two distinct outcomes, provides a potential mechanism by which individuals may come to be generally more influenced by events of one class than another. This finding may be relevant to clinical questions. In the case of depression, patients have been shown to be more influenced by negative events, for example tending to remember more negative than positive events (*Bradley et al., 1995*), attend to negative more than positive events (*Gotlib et al., 2004*) and learn more from negative and less from positive outcomes (*Eshel and Roiser, 2010*). As the negative biases described above are believed to be causally related to symptoms of depression (*Mathews and MacLeod, 2005*), and interventions designed to alter negative biases can reduce symptoms (*Browning et al., 2012*; *NICE, 2009*), these results raise the possibility that novel interventions which target exstimated information content may act to alter negative affective biases and thus reduce symptoms of the illness. Of course, identifying potential targets for treatment and showing that they may be altered experimentally as done in this paper is only the first step in the development of new treatments. The next step, analogous to a phase 2a study in drug development (*Ciociola et al., 2014*), is to assess the initial efficacy of a potential intervention which engages the target in a clinical population. A study designed to do this is currently underway using the volatility manipulation described in this paper (study identifier NCT02913898).

While the results of the current study provide evidence that the information content of different events can be estimated in parallel during learning, the level of abstraction at which these estimates

function is not clear. For example, does the task used in the current study alter the estimated information content of all positive and negative outcomes, or just those used in the task? This question is relevant to the potential clinical application of interventions which modify this estimate, as the affective biases associated with emotional disorders are seen across a wide variety of contexts (*Mathews and MacLeod, 2005*) and thus an intervention which modifies one particular instance of bias is unlikely to be useful therapeutically. It is clearly unlikely that completing a single block of a learning task, as done in the current study, will produce a broad and generalised alteration of the estimated information content of all outcomes. Rather, in order to test the degree to which alteration of estimated information content generalises it will be necessary to repeatedly expose participants to situations in which one class of outcome (e.g. positive) is more informative and then measure whether this alters learning performance in separate tasks and, ultimately, whether it impacts on clinical symptoms. The intervention used in the ongoing clinical study described above involves repeatedly completing the 'positive volatile' block from the current study over the course of two weeks (see; *Browning et al., 2012* for a similar design) which will provide an initial assessment of this question.

The information content of an outcome is not solely a function of the volatility of its occurrence. Other factors, such as the strength of the association between a stimulus, or action, and the subsequent outcome, sometimes called the 'expected uncertainty' (*Yu and Dayan, 2005*) of the association, will also influence how informative the outcome is. Outcomes in the learning task reported in this paper vary in terms of both volatility and expected uncertainty, with both of these factors predicted to influence learning rate in the same direction (i.e. both factors should increase learning rate in the volatile blocks). An additional experiment (see *Figure 2—figure supplement 2*) in which volatility was kept constant but expected uncertainty varied found no effect on learning rate suggesting that the current findings were likely to be due to the effects of volatility rather than expected uncertainty. However, it would be interesting in future studies to explore whether it was possible to use manipulations of expected uncertainty, in the same way that volatility is used in this study, to induce a preference for positive over negative events. This may provide an alternative approach to engaging and altering expected information content than the volatility based effect reported here.

The current study demonstrates that human learners maintain separable estimates of the information content of distinct positive and negative outcomes and provides an initial proof of principle as to how these estimates may be modified. The study illustrates a little explored application of computational techniques in cognitive neuroscience; they may be used to identify potential novel treatment targets and by so doing spur the development of new and more effective treatments.

## Materials and methods

### Participants

30 English-speaking, individuals aged between 18 and 65 were recruited from the local community via advertisements. The number of participants recruited for the current cohort was selected to provide >95% power of detecting a similar effect size as that reported in a previous study in which a volatility manipulation was used to influence learning rate (*Browning et al., 2015*). Potential participants who were currently on a psychotropic medication or who had a history of neurological disorders were excluded from the study.

### General procedure

The study involved a single experimental session during which participants completed a novel learning task (described below) as well as standard questionnaire measures of depression (Quick Inventory of Depressive Symptoms, QIDS [*Rush et al., 2003*]) and anxiety (Spielberger State-Trait Anxiety Inventory, trait subscale, STAI [*Spielberger et al., 1983*]) symptoms. The study was approved by the University of Oxford Central Research Ethics Committee. Written informed consent was obtained from all participants, in accordance with the Declaration of Helsinki.

### The information bias learning task

The information bias learning task (*Figure 1*) was adapted from a structurally similar learning task previously reported in the literature (*Behrens et al., 2007*; *Browning et al., 2015*). On each trial of

the task participants were presented with two abstract shapes (letters selected from the Agathodaimon font) and chose the shape which they believed would result in the best outcome. On each trial, the win and loss outcomes were independently positioned (both had 15 p magnitude) such that a particular shape could be associated with one, both or neither of the win and loss outcomes (*Figure 1C*). As the two outcomes were independent participants had to separately learn the likely location of the win and the loss in the current trial. This learning was driven by the outcomes of previous trials and was used by participants to determine the most advantageous shape to choose on the current trial. Throughout the task the number and type of stimuli displayed during each phase of the trials was kept constant (*Figure 1a*) in order to minimise variations in luminance between trials.

In total, the participants completed three blocks of 80 trials each, with a rest session between blocks. The same two shapes were used for all trials within a block, with different shapes being used between blocks. The outcome schedules were determined such that the probability that wins and losses were associated with shape A within a block always averaged 50%. In the volatile blocks the association between shape A and the outcome changed from 15% to 85% and back again in runs ranging from 14 to 30 trials. As described in the introduction, outcomes in the volatile blocks were more useful when predicting future outcomes, making them 'informative', whereas in the stable blocks outcome probabilities were fixed at 50%, making the outcomes 'uninformative' in terms of predicting future trials (*Figure 1B*). In the first block of the task, both outcomes were volatile (informative), whereas in blocks 2 and 3 only one of the outcomes was volatile (informative) with the other being stable (uninformative). See *Figure 2—figure supplement 2* for results from an additional task in which volatility was kept constant, while the strength of the association between stimuli and outcomes (i.e. expected uncertainty) was varied. The order in which blocks 2 and 3 were completed was counterbalanced across participants. Participants were paid all the money they had collected in the task, in addition to a £10 baseline payment. Choice data from the task was analysed by fitting a behavioural model which is described and compared to alternative models below.

The task was presented on a VGA monitor connected to a laptop computer running Presentation software version 18.3 (Neurobehavioural Systems, Berkeley, CA). Participants' heads were stabilised using a head-and-chin rest placed 70 cm from the screen on which an eye tracking system was mounted (Eyelink 1000 Plus; SR Research, Ottawa, Canada). The eye tracking device was configured to record the coordinates of both of the eyes and pupil area at a rate of 500 Hz. The abstract shapes of the learning task were drawn on either side of a fixation cross which marked the middle of the screen and were offset by around 7° visual angle. The two outcomes (win and loss) were displayed on the screen in randomised order for a jittered interval of 2–6 (mean 4) seconds. Auditory stimuli lasting 0.7 s were played when participants received a win ('chi-ching' sound) or loss (error buzz). Participants' accumulated total winnings was displayed under the fixation cross and was updated at the beginning of the subsequent trial.

## Behavioural model used in analysis of the learning task

The primary measure of interest in the learning task is the learning rate for wins and for losses in each of the three blocks. A simple behavioural model, based on that employed in related tasks (*Behrens et al., 2007*; *Browning et al., 2015*) was used to estimate learning rate. This model first estimated the separate probabilities that the win and loss would be associated with shape 'A' using a Rescorla-Wagner learning rule (*Rescorla and Wagner, 1972*):

$$rwin_{(i+1)} = rwin_{(i)} + \alpha win * \left( winout_{(i)} - rwin_{(i)} \right)$$

$$rloss_{(i+1)} = rloss_{(i)} + \alpha loss * \left( lossout_{(i)} - rloss_{(i)} \right)$$

In these equations $rwin_{(i)}$, which was initialised at 0.5, is the estimated probability that the win will be associated with shape 'A' on trial i (NB the probability that the win is associated with shape 'B' is 1-$rwin_{(i)}$), $winout_{(i)}$ is a variable coding for whether the win was associated with shape 'A' (in which case the variable has a value of 1) or shape 'B' (giving a value of 0) and $\alpha win$ is a free parameter, the learning rate for the wins. $rloss_{(i)}$, $lossout_{(i)}$ and $\alpha loss$ are the same variables for the loss outcome. These estimated outcome probabilities were then transformed into a single choice probability using a soft max function:

$$PchoiceA_{(i)} = \frac{1}{1 + exp^{-\left(\beta win * rwin_{(i)} - \beta loss * rloss_{(i)}\right)}}$$

Where $PchoiceA_{(i)}$ is the probability of choosing shape 'A' on trial i, and $\beta win$ and $\beta loss$ are inverse decision temperatures for wins and losses, respectively. The four free-parameters of this model (learning rates and inverse temperatures for wins and losses) were estimated separately for each task block and each participant by calculating the full joint posterior probability of the parameters, given participants' choices, and then deriving the expected value of each parameter from their marginalised probability distributions (*Behrens et al., 2007*; *Browning et al., 2015*). Choice data from the first 10 trials of each block was not used when estimating the parameters as these trials were excluded from the pupil analysis (due to initial pupil adaption) (*Browning et al., 2015*; *Nassar et al., 2012*). Apart from the main behavioural analysis reported in *Figure 2*, the first block of the task in which both wins and losses had a volatile outcome probability schedule were excluded from subsequent behavioural and pupil analysis. This first block of the task was designed to acclimatise participants to the task.

## Alternative behavioural models and model selection

The behavioural model used in this study (Referred to as model 1 below) was developed based on the models used in previous studies in which volatility is manipulated (*Behrens et al., 2008*, *Behrens et al., 2007*; *Browning et al., 2015*). However, it is possible that this model does not provide the best fit to participant choice data. In order to assess this possibility we compared the fit of this model against a range of comparator models using the Bayesian Information Criteria (BIC) metric, which includes a penalty term for model complexity.

Model 2: It is possible for participants to perform our task without learning the independent probability of the win and loss outcomes, but rather by taking a model-free (*Daw et al., 2011*) approach in which the overall value of each shape was learned.

$$v^A_{(i+1)} = v^A_{(i)} + \alpha value * \left(out_{(i)} - v^A_{(i)}\right)$$

Here the value of shape A ($v^A$) initiates at 0 on trial 1, and is updated on every trial based on the joint outcome (i.e. the win – loss for that shape) of the trial $\left(out_{(i)}\right)$, which can be $-1$, 0 or 1 with a single learning rate ($\alpha value$). The estimated relative values of the 2 shapes were then transformed into a choice probability using a softmax function with a single inverse temperature parameter.

Model 3: An alternative approach, described by Behrens and colleagues (*Behrens et al., 2007*) estimates trialwise volatility within a fully Bayesian framework. For this model we used Behrens' Bayesian learner to independently estimate the expected probabilities of the win and loss outcomes during the task (note that there are no free parameters for this learner). These estimates were then combined using the same selector model described in the main text with two inverse temperature parameters.

Model 4: This was a slightly simpler version of Model 1 in that it employed only a single inverse temperature parameter allowing assessment of the degree to which using 2 such parameters influenced model fit.

Model 5: Finally, we tested a slightly more complex version of Model 4 by including a risk parameter $\gamma$, as used in previous studies, which modulates the estimated probabilities of wins and losses in a non-linear way. Risk parameters have been shown to account for non-normative aspects of human choice (*Browning et al., 2015*; *Prelec, 1998*), particularly when outcome probabilities are particularly high or low:

$$\tilde{rwin}_{(i)} = 2^{-\left(-log_2\left(rwin_{(i)}\right)^\gamma\right)}$$
$$\tilde{rloss}_{(i)} = 2^{-\left(-log_2\left(rloss_{(i)}\right)^\gamma\right)}$$

A summary of the five models can be found in *Table 2* below:

All models were fitted to participant data using the same procedure described in the main paper. BIC scores for each model are illustrated in *Figure 5* below (note that lower scores indicate a better fit). As can be seen the model reported in the main paper (Model 1) fits the data best. The single

**Table 2.** Description of Comparator Models

| Model name | Number of learning rate parameters | Number of inverse temperature parameters | Notes |
| --- | --- | --- | --- |
| 1. | 2 | 2 | Model used in paper |
| 2. | 1 | 1 | Model-free learner |
| 3. | 0 | 2 | Bayesian learner |
| 4. | 2 | 1 | Single inverse temperature model |
| 5. | 2 | 1 | Additional risk parameter |

DOI: https://doi.org/10.7554/eLife.27879.014

inverse temperature model (Model 4) performs almost as well, with the other models performing less well.

## Pupilometry data preprocessing

Blinks were identified using the Eyelink system's built in filter and were then removed from the data. Missing data points (including blinks) were linearly interpolated. The resulting trace was subjected to a low pass Butterworth filter with a cut-off of 3.75 Hz and then z transformed across the session (*Browning et al., 2015*; *Nassar et al., 2012*). The pupil response to the win and the loss outcomes were extracted separately from each trial, using a time window based on the presentation of the outcomes. This included a 1 s baseline period before the presentation of the outcome, and a 6 s period following outcome presentation. Baseline correction was performed by subtracting the mean pupil size during the 1 s baseline period prior to the presentation of each outcome, from each time point in the post outcome period. Individual trials were excluded from the pupilometry analysis if more than 50% of the data from the outcome period had been interpolated (mean = 7% of trials) (*Browning et al., 2015*). One participant was excluded from the pupilometry analysis as more than 99% of their trials were excluded on this basis. The first 10 trials from each block were not used in the analysis as initial pupil adaption can occur in response to luminance changes in this period (*Browning et al., 2015*; *Nassar et al., 2012*). The preprocessing resulted in two sets of timeseries per participant, one set containing pupil dilation data for each included trial when the win outcomes were displayed and the other when the loss outcomes were displayed. A difference timeseries, calculated as the mean pupil response when the outcomes appears on the chosen versus unchosen shape in each block was then calculated which allowed for assessment of how the volatility of a specific outcome influenced dilation in response to receiving vs. not receiving that outcome (See *Figure 3—figure supplement 2* for a complementary regression analysis of this data).

Preprocessing resulted in difference timeseries of pupil dilation data which represented the differential pupil dilation occurring during trials when the outcome (win or loss) was received relative to when it was not received over the six seconds after presentation of the outcomes. These timeseries were binned into 1 s bins to facilitate analysis.

## Data analysis

Parameters derived from the computational models were transformed before analysis so that they were on the infinite real line (an inverse logit transform was used for learning rates and a log transform for inverse temperatures). Where possible figures illustrate non-transformed parameters for ease of interpretation. The effect of the volatility manipulation on these transformed parameters was tested using a repeated measures ANOVA of data derived from the last two task blocks (i.e. when volatility was manipulated). In this ANOVA block volatility (win volatile block, loss volatile block) and parameter valence (wins, losses) were within subject factors and block order (win volatile first, loss volatile first) was a between subject factor. The critical term of this analysis is the block volatility x parameter valence interaction which tests for a differential effect of the volatility manipulation on the win and loss parameters.

The binned pupil timeseries data was analysed using a repeated measures ANOVA with time bin (1–6 s), block volatility (win volatile, loss volatile) and valence (wins, losses) as within subject factors and block order as a between subject factor. Again a block volatility x valence interaction tests for a differential effect of the volatility manipulation on the pupil dilation in response to wins vs. losses.

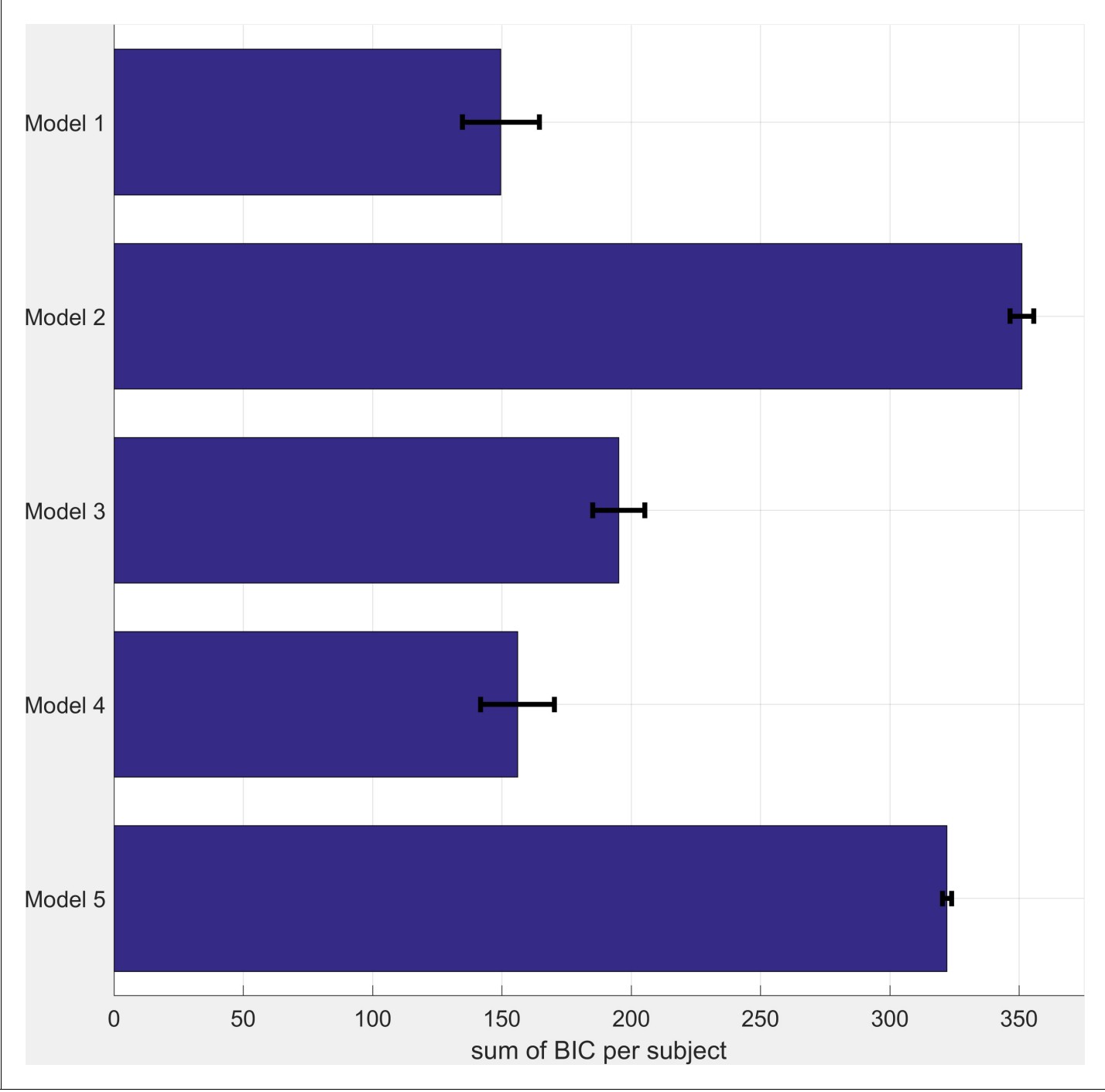

**Figure 5.** BIC Scores for Comparator Models (see table S1 for model descriptions). Smaller BIC scores indicate a better model fit. BIC scores were calculated as the sum across all three task blocks. Bars represent mean (SEM) of the scores across participants.

DOI: https://doi.org/10.7554/eLife.27879.015

We tested whether the volatility effect was larger for loss than win outcomes using a similar ANOVA in which block volatility was replaced by 'outcome volatility' (i.e. outcome volatility is high when the volatility of a given outcome, wins or losses, is high). In order to perform between subject correlations of the pupilometry data the mean relative dilation across the entire six second outcome period was also calculated for each participant and each block. In all analyses significant interactions were followed up by standard post-hoc tests.

## Acknowledgements

This study was funded by a MRC Clinician Scientist Fellowship awarded to MB (MR/N008103/1). MB has received travel expenses from Lundbeck for attending conferences. EP declares no potential conflict of interest.

## Additional information

### Competing interests

Michael Browning: Received travel expenses from Lundbeck for attending conferences. The other author declares that no competing interests exist.

### Funding

| Funder | Grant reference number | Author |
| --- | --- | --- |
| Medical Research Council | MR/N008103/1 | Michael Browning |

The funders had no role in study design, data collection and interpretation, or the decision to submit the work for publication.

### Author contributions

Erdem Pulcu, Formal analysis, Writing—original draft; Michael Browning, Conceptualization, Formal analysis, Supervision, Funding acquisition, Writing—original draft

### Author ORCIDs

Michael Browning  http://orcid.org/0000-0001-9108-3144

### Ethics

Human subjects: All participants provided written informed consent. The study was reviewed and approved by the Medical Sciences Interdepartmental Research Ethics Committee of Oxford University (ref number MSD-IDREC-C1-2014-216).

### Decision letter and Author response

Decision letter https://doi.org/10.7554/eLife.27879.017
Author response https://doi.org/10.7554/eLife.27879.018

## Additional files

### Supplementary files

• Transparent reporting form
DOI: https://doi.org/10.7554/eLife.27879.016

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
