## [Decision Letter]

Thank you for submitting your article "Affective Bias as a Rational Response to the Statistics of Rewards and Punishments" for consideration by *eLife*. Your article has been reviewed by two peer reviewers, and the evaluation has been overseen by Reviewing Editor Michael Frank and Sabine Kastner as the Senior Editor. The reviewers have opted to remain anonymous.

The reviewers have discussed the reviews with one another and the Reviewing Editor has drafted this decision to help you prepare a revised submission.

Summary:

Pulcu and Browning lay the groundwork to examine affective biases in depression in a novel and very informative manner. Specifically, they suggest that the affective bias towards negative events seen in depression across many different paradigms reflects a tendency to judge negative events as more "informative". They designed an experiment in which the valence and the informativeness of outcomes were independent: at times, positive outcomes were more informative, at times negative ones. What they find is that individuals accurately infer the informativeness of the different events, and update their predictions accordingly. When wins are informative (because the cue indicating wins can change), then wins lead to stronger behavioural adaptation. The same is true for losses. Given that learners should generally put more weight on observations that are more predictive or informative, the paper proposes that overweighting of negative events might reflect an overestimate of their informativeness. Strikingly, they (approximately) replicate their previously reported association of pupil dilation with volatility, but find this relationship to be valence-dependent, as it was true of negative outcomes, but not for wins.

Essential revisions:

The paper is a pleasure to read. It is well written, the methods are beyond reproach, and the combination of computational modelling with behaviour, neurophysiological measures and psychopathology is to be commended. However, there were a few important issues for which we would like clarification in order to proceed.

1) Both reviewers noted that the remarkable finding (Figure 3) of a valence-dependent effect of volatility on pupil dilation did not seem to be tested statistically. There was a significant effect for losses and not gains, but we did not see a direct contrast between the two. If we've correctly understood the reported interaction in the subsection “Does Activity of the Central NE System, as Estimated by Pupil Dilation, Track the Volatility of Positive and Negative Outcomes?”, it does not test specifically for gain/loss asymmetry but is rather analogous to the 2nd and 3rd groups of bars in Figure 2. The volatility factor is "block volatility" (wins-volatile versus losses-volatile). So if there were merely an overall effect of outcome volatility, it would come through here as an interaction of block type x valence. This could probably be dealt with by setting up the factors as "outcome volatility" and "outcome valence" rather than "block volatility" and "outcome valence", and reporting the interaction. Secondarily, the conclusions section asserts that pupillary response was larger overall for losses (Discussion, third paragraph) but I don't see where this was tested.

2) Similarly, the finding that pupil dilation predicts learning rate for losses and not gains does not itself show that the correlation is stronger for one than the other (this can be tested e.g. via Fisher r-to-z transformation). Moreover, it is somewhat unclear what it means that the behavioral effect of volatility is just as strong in win vs. loss conditions but that the pupil doesn't care about wins – can the authors speculate about what this means for separate mechanisms of learning rate adjustment for non-aversive outcomes? (and how does this fit with previous data linking pupil dilation to learning).

3) Both reviewers also noted that the framing does not quite match up with the main results. The Abstract concludes, "Humans maintain independent estimates of the information content of positive and negative outcomes". This implies people might estimate the informativeness of positive and negative outcomes as general categories, which would go a long way toward explaining affective bias. But what the paper actually shows is quite a bit narrower, that people can track the relative informativeness of two individual outcomes. It's not clear if valence has a privileged role compared to any other feature that might individuate the outcomes (and perhaps there is even a sensory confound in differences between auditory and visual characteristics of the outcomes), nor is it clear if people generalize informativeness estimates across events of the same valence. So it would be appropriate to reduce the breadth of this conclusion (which shows up in a number of places in the paper, including the next-to-last sentence of the Abstract and the first paragraph of the Discussion) and to discuss or qualify this potential limitation.

4) Relatedly, it is not really obvious that these wins and losses engage much 'affect', and indeed it is unclear whether anything lasting has really changed within the individuals other than that they performed the task well and correctly inferred the task structure. To support the statement about such a procedure changing affective biases, it would probably need some independent measure of that change, e.g. a change in one of the measures of negative affective bias typically associated with depression, e.g. memory or attention or learning on a separate task. An alternative would be to provide some psychopathological correlate, e.g. with rumination or dysfunctional attitudes, neuroticism or so, akin to how they were able to show an association with STAI in their previous paper. Given the broad nature of the issue there really is a long list of potential measures one could choose from. In the Materials and methods, the QIDS and the STAI were acquired – maybe these could be used?

5) For the control experiment described in Figure 2—figure supplement 2, it would be helpful to have additional information about how noise and volatility were parameterized, and what parameters were used. In other words, what was the generative model for the outcome schedules? I also didn't understand why this was referred to as a control experiment. It seems instead like a second test of the main hypothesis: if people adjust learning based on outcome informativeness, then noise level should make a difference. Given that it doesn't, it seems like this should be interpreted as a caveat or limitation on adaptive learning (rather than as a control for an alternative explanation).

[Editors' note: further revisions were requested prior to acceptance, as described below.]

Thank you for resubmitting your work entitled "Affective Bias as a Rational Response to the Statistics of Rewards and Punishments" for further consideration at *eLife*. Your revised article has been favorably evaluated by Sabine Kastner as the Senior Editor, Michael Frank as the Reviewing editor, and two anonymous reviewers.

The reviewers both agreed that the manuscript has been improved but there are some remaining issues that need to be addressed before acceptance, as outlined below:

In particular, reviewer 1 points out that the framing of the manuscript in terms of depression – given the lack of association with clinical measures – continues to miss out on an opportunity to highlight what the findings speak to more directly. In addition to the reviewer comments below, in the consultation session amongst reviewers and editor it was suggested that you could emphasize any or all of the following aspects:

The study shows that people are able to adjust learning rates in a way that is more clearly indicative of the latent causes compared to previous studies in this domain (i.e. it is feasible that the typical volatility learning rate association could have arisen from a generalized response to arousal or overall changeability of the world leading to more learning writ large, but here adjustment is clearly more sophisticated and rational). Other aspects that the study is relevant to mentioned by reviewers are the known neurobiology of learning from rewards and losses more generally, loss aversion, unified scalar value codes, the relative roles of dopamine, acetylcholine and noradrenaline in this sort of task. All of these are entirely independent of any depression/psychopathology.

If you feel strongly to maintain the current framing in terms of depression given that this clearly was your motivation in the first place, that would be fine as such normative approaches to prominent and important issues in psychopathology are indeed very promising and rare. But we felt that discussing/ motivating the above and below issues could further improve your paper.

Reviewer #1:

Pulcu and Browning have, in my view, responded clearly to all requests.

What is clear and convincing now is the extent to which negative outcomes differ from positive ones, and this per se is a novel and important contribution.

The main caveat, in my view, remains the interpretation in terms of depression.

First, no relationship to any symptoms of depression is demonstrated or examined despite such symptoms having been measured at baseline with two different instruments. Second, in depression, it seems like the judgement aboutinformativeness is maladaptive rather than adaptive, which raises questionsabout what such an adaptive paradigm says about depression. Third, hereinformativeness is closely related to malleability, but in a way which is notobviously the case for depression. Here more volatile outcomes judged are asmore informative and hence given more weight in terms of influencing thefollowing choices, but they are also more malleable. The notion of maladaptivecognitive schemas in depression suggest a different relationship, whereby 'moreinformative' is also linked to 'less malleable'.

In defence of the authors, they do state clearly that this intervention is likely insufficient to change symptoms, and that longer training sessions would be required before any change should be expected. They also make it clear that the processes to be examined here are explicitly aimed at change and intervention, and not necessarily at measuring existing biases per se. However, the weakness of the manuscript as it is written is that it continues to be focused strongly on depression, even though the results don't yet speak to this at all. Instead, they do speak in interesting manners about learning from rewards and losses. The framing and discussion in terms of this literature is, in my view, a missed opportunity, and as such I continue to think that the paper is framed somewhat unfortunately and would benefit from being framed differently.

Reviewer #2:

The authors have done a nice job with the revised manuscript. All the additional information I requested is now included, and the clarifications and added points of discussion are helpful and on-point. In particular, I appreciate the clear statement that the present study's objective was to test a candidate mechanism, not to measure effects of the experimental manipulation on clinical symptoms. The additional information about the ongoing clinical study is helpful in clarifying how the present study fits within a broader clinical research agenda.

---

## [Author Response]

Essential revisions: The paper is a pleasure to read. It is well written, the methods are beyond reproach, and the combination of computational modelling with behaviour, neurophysiological measures and psychopathology is to be commended. However, there were a few important issues for which we would like clarification in order to proceed.1) Both reviewers noted that the remarkable finding (Figure 3) of a valence-dependent effect of volatility on pupil dilation did not seem to be tested statistically. There was a significant effect for losses and not gains, but we did not see a direct contrast between the two. If we've correctly understood the reported interaction in the subsection “Does Activity of the Central NE System, as Estimated by Pupil Dilation, Track the Volatility of Positive and Negative Outcomes?”, it does not test specifically for gain/loss asymmetry but is rather analogous to the 2nd and 3rd groups of bars in Figure 2. The volatility factor is "block volatility" (wins-volatile versus losses-volatile). So if there were merely an overall effect of outcome volatility, it would come through here as an interaction of block type x valence. This could probably be dealt with by setting up the factors as "outcome volatility" and "outcome valence" rather than "block volatility" and "outcome valence", and reporting the interaction. Secondarily, the conclusions section asserts that pupillary response was larger overall for losses (Discussion, third paragraph) but I don't see where this was tested.

As suggested we have added an additional analysis to the manuscript which confirms that the difference in magnitude of the volatility effect for the win and loss outcomes was statistically significant:

“Indeed a direct statistical comparison of the size of the volatility effect between the positive and negative outcomes indicated a greater effect of volatility on the negative relative to positive outcomes (outcome volatility x valance; F(1,27)=4.34, p=0.047).”

Please note that the main block volatility x valence effect reported in the original paper should have been p=0.02 rather than p=0.04, we have corrected this and apologise for the error. A brief description of the above analysis has been added to the Materials and methods section of the paper:

“We tested whether the volatility effect was larger for loss than win outcomes using a similar ANOVA in which block volatility was replaced by “outcome volatility” (i.e. outcome volatility is high when the volatility of a given outcome, wins or losses, is high).”

We have also added in the requested analysis confirming that pupil dilation to losses was greater than to wins:

“This effect was seen on the background of a generally greater pupil dilation to receipt of a loss relative to a win (main effect of valence; F(1,27)=16.7, p<0.001).”

2) Similarly, the finding that pupil dilation predicts learning rate for losses and not gains does not itself show that the correlation is stronger for one than the other (this can be tested e.g. via Fisher r-to-z transformation). Moreover, it is somewhat unclear what it means that the behavioral effect of volatility is just as strong in win vs. loss conditions but that the pupil doesn't care about wins – can the authors speculate about what this means for separate mechanisms of learning rate adjustment for non-aversive outcomes? (and how does this fit with previous data linking pupil dilation to learning).

We have now conducted the Fisher r-to-z transformation analysis which confirms that the correlations are significantly greater for losses than wins:

“This correlation was significantly greater for losses than for wins (Fisher r-to-z transformation z=2.27, p=0.02).”

We agree with the reviewers that the interpretation of this difference is not straightforward. While we also agree that the results raise the possibility of specificity in the pupillary response to negative vs. positive outcomes we think that caution is warranted in this interpretation. We have elaborated our thoughts on this in a thoroughly revised section in the Discussion of the paper (fifth and sixth paragraphs). In summary, we suggest a) that the previous work describing increased pupil dilation for volatile relative to stable rewards suggests that the central NE system does in fact respond to the volatility of positive outcomes, b) that the smaller effect of volatility for positive outcomes in the current study may relate to fact that the volatility signal in the current study is conditioned on receipt of the outcome and the overall dilation to rewards is smaller than that to losses and finally c) this proposal may be tested in future work by varying the salience (e.g. by varying the magnitude) of positive vs. negative outcomes which would have the effect of varying the dilation to the outcomes or by testing the impact of pharmacological manipulations on win vs. loss learning rate:

“The effect of outcome volatility on pupil dilation in the current study was significantly greater for loss than win outcomes with the correlation between this signal and behaviour also significantly greater for losses. […] Specifically, a greater impact of a pharmacological manipulation on learning rates for losses than wins would provide experimental evidence for a preferential role for this system in estimates of the information content of losses.”

3) Both reviewers also noted that the framing does not quite match up with the main results. The Abstract concludes, "Humans maintain independent estimates of the information content of positive and negative outcomes". This implies people might estimate the informativeness of positive and negative outcomes as general categories, which would go a long way toward explaining affective bias. But what the paper actually shows is quite a bit narrower, that people can track the relative informativeness of two individual outcomes. It's not clear if valence has a privileged role compared to any other feature that might individuate the outcomes (and perhaps there is even a sensory confound in differences between auditory and visual characteristics of the outcomes), nor is it clear if people generalize informativeness estimates across events of the same valence. So it would be appropriate to reduce the breadth of this conclusion (which shows up in a number of places in the paper, including the next-to-last sentence of the Abstract and the first paragraph of the Discussion) and to discuss or qualify this potential limitation.

We agree with the reviewers that there is an interesting outstanding question related to the level of abstraction at which the observed effect operates. We have amended the description of our findings throughout the study to highlight that participants learned differently from “distinct” outcomes and have added a new section to the Discussion in which this question is covered. As this issue is closely linked to the next point raised by the reviewers, this novel section is reproduced after point 4 below.

4) Relatedly, it is not really obvious that these wins and losses engage much 'affect', and indeed it is unclear whether anything lasting has really changed within the individuals other than that they performed the task well and correctly inferred the task structure. To support the statement about such a procedure changing affective biases, it would probably need some independent measure of that change, e.g. a change in one of the measures of negative affective bias typically associated with depression, e.g. memory or attention or learning on a separate task. An alternative would be to provide some psychopathological correlate, e.g. with rumination or dysfunctional attitudes, neuroticism or so, akin to how they were able to show an association with STAI in their previous paper. Given the broad nature of the issue there really is a long list of potential measures one could choose from. In the Materials and methods, the QIDS and the STAI were acquired – maybe these could be used?

We thank the reviewers for highlighting this issue and agree that the degree to which the effect observed in this study generalises to different situations (or even to clinical symptoms) is not clear. The focus of our work was to test basic mechanistic questions (do people actually maintain separate estimates of the information content of different outcomes? Can we modify these and are they linked to central NE function?) which are potentially relevant to why affective biases may develop. As a result of this focus our study was set up using a single session, within subject design. All participants therefore completed a single “positive volatile” block and a single “negative volatile” block. As it was implausible that completion of a single block of a learning task would significantly alter subjective symptoms of depression we did not collect these after completion of each of the separate blocks (NB subjective symptoms of depression generally change over the course of weeks of treatment). We have highlighted the question of effect generalisation in a new Discussion section in which we describe the outstanding questions (also related to point 3 above) and how they might be answered using different study designs:

“The results of the current study provide evidence that the information content of different events can be estimated in parallel during learning, however the level of abstraction at which these estimates function is not clear. […] The intervention used in the ongoing clinical study described above involves repeatedly completing the “positive volatile” block from the current study over the course of two weeks (see; Browning et al., 2012 for a similar design) which will provide an initial assessment of this question.”

5) For the control experiment described in Figure 2—figure supplement 2, it would be helpful to have additional information about how noise and volatility were parameterized, and what parameters were used. In other words, what was the generative model for the outcome schedules? I also didn't understand why this was referred to as a control experiment. It seems instead like a second test of the main hypothesis: if people adjust learning based on outcome informativeness, then noise level should make a difference. Given that it doesn't, it seems like this should be interpreted as a caveat or limitation on adaptive learning (rather than as a control for an alternative explanation).

We have added in details of the generative process used to determine magnitude size in this task (added to the legend of Figure 2—figure supplement 2):

“This design allowed us to present participants with schedules in which the volatility (i.e. unexpected uncertainty) of win and loss magnitudes was constant (three change points occurred per block) but the noise (expected uncertainty) varied (Panel b; the standard deviation of the magnitudes was 17.5 for the high noise outcomes and 5 for the low noise outcomes).”

We had added this task to tease apart the effects of expected and unexpected uncertainty reported in the main paper, but agree that perhaps calling this a “control” task is confusing. We have now renamed it descriptively as the “magnitude” task which is described in the main paper as an “additional” task.

We have also added in an additional discussion about the results from this task and how it might be interpreted in light of the paper by Nassar and colleagues which reported an effect of expected uncertainty on reward learning using a very different task to that reported here. Given this finding we don’t think the result of our task can be taken as strong evidence for a limitation to the adaptive learning account, but rather that our task is more sensitive to differences in unexpected than expected uncertainty. This additional discussion has been added to the legend for Figure 2—figure supplement 2:

“Interestingly a previous study (Nassar et al., 2012) described a learning tasks in which a normative effect of outcome noise was seen (i.e. a higher learning rate was used by participants when the outcome had lower noise). […] Regardless of the exact reason for the lack of effect of noise in the magnitude task, it suggests that the effect described in the main paper is likely to be driven by an effect of unexpected rather than expected uncertainty.”

[Editors' note: further revisions were requested prior to acceptance, as described below.]

The reviewers both agreed that the manuscript has been improved but there are some remaining issues that need to be addressed before acceptance, as outlined below:In particular, reviewer 1 points out that the framing of the manuscript in terms of depression – given the lack of association with clinical measures – continues to miss out on an opportunity to highlight what the findings speak to more directly. In addition to the reviewer comments below, in the consultation session amongst reviewers and editor it was suggested that you could emphasize any or all of the following aspects:The study shows that people are able to adjust learning rates in a way that is more clearly indicative of the latent causes compared to previous studies in this domain (i.e. it is feasible that the typical volatility learning rate association could have arisen from a generalized response to arousal or overall changeability of the world leading to more learning writ large, but here adjustment is clearly more sophisticated and rational). Other aspects that the study is relevant to mentioned by reviewers are the known neurobiology of learning from rewards and losses more generally, loss aversion, unified scalar value codes, the relative roles of dopamine, acetylcholine and noradrenaline in this sort of task. All of these are entirely independent of any depression/psychopathology.If you feel strongly to maintain the current framing in terms of depression given that this clearly was your motivation in the first place, that would be fine as such normative approaches to prominent and important issues in psychopathology are indeed very promising and rare. But we felt that discussing/ motivating the above and below issues could further improve your paper.Reviewer #1:Pulcu and Browning have, in my view, responded clearly to all requests.What is clear and convincing now is the extent to which negative outcomes differ from positive ones, and this per se is a novel and important contribution.The main caveat, in my view, remains the interpretation in terms of depression.First, no relationship to any symptoms of depression is demonstrated or examined despite such symptoms having been measured at baseline with two different instruments. Second, in depression, it seems like the judgement aboutinformativeness is maladaptive rather than adaptive, which raises questionsabout what such an adaptive paradigm says about depression. Third, hereinformativeness is closely related to malleability, but in a way which is notobviously the case for depression. Here more volatile outcomes judged are asmore informative and hence given more weight in terms of influencing thefollowing choices, but they are also more malleable. The notion of maladaptivecognitive schemas in depression suggest a different relationship, whereby 'moreinformative' is also linked to 'less malleable'.In defence of the authors, they do state clearly that this intervention is likely insufficient to change symptoms, and that longer training sessions would be required before any change should be expected. They also make it clear that the processes to be examined here are explicitly aimed at change and intervention, and not necessarily at measuring existing biases per se. However, the weakness of the manuscript as it is written is that it continues to be focused strongly on depression, even though the results don't yet speak to this at all. Instead, they do speak in interesting manners about learning from rewards and losses. The framing and discussion in terms of this literature is, in my view, a missed opportunity, and as such I continue to think that the paper is framed somewhat unfortunately and would benefit from being framed differently.Reviewer #2:The authors have done a nice job with the revised manuscript. All the additional information I requested is now included, and the clarifications and added points of discussion are helpful and on-point. In particular, I appreciate the clear statement that the present study's objective was to test a candidate mechanism, not to measure effects of the experimental manipulation on clinical symptoms. The additional information about the ongoing clinical study is helpful in clarifying how the present study fits within a broader clinical research agenda.

We thank the reviewers and editor for this assessment and agree with them that the weakest aspect in the previous manuscript was our suggestion of a definite link between our findings and the symptoms of depression (which we did not demonstrate). We have thoroughly revised the Introduction and Discussion section of the manuscript in order to make the following changes:

1) General framing of the study: We have changed the framing of the study so that it is now introduced in terms of affective bias (i.e. the tendency to learn differently from rewards relative to losses) rather than as directly relevant to depression. We have completely removed discussion of depression in the Introduction (and Abstract) with the exception of noting that it is associated with negative affective biases and that understanding biases is essential in developing new treatments (as noted, this was the motivating factor for the study). We have reordered the Discussion to highlight the mechanistic sections relative to the “clinical application” sections. We have also removed completely the sections in which we suggested that the volatility effect described in the paper provides a compelling account of the development of negative bias in depression (we agree with reviewer 1 that it cannot account for all elements of the negative biases seen in depression). The sections of the Discussion which do cover potential clinical applications of the current findings are now focused only on the development of new treatments (and using computational approaches to achieve this).

2) Additional sections added to the Introduction/Discussion sections: In response to the editorial suggestion that a number of more mechanistic issues would provide a better framing of the study we have made the following changes:

a) We have added additional text to both the Introduction and Discussion sections highlighting that previous volatility findings may have arisen due to learners estimating general environmental volatility, whereas our results demonstrate a more sophisticated learning system which requires outcome specific estimates of volatility.

b) We have discussed the asymmetry of the pupilometry data (i.e. a larger overall response as well as greater volatility signal to losses relative to gains) with reference to similar findings in the loss avoidance literature.

Overall, we hope that we have improved the paper by refocusing on the mechanistic issues which are most closely linked to our results while maintaining those treatment relevant aspects which speak to future clinical application.